# Innate Immune Signaling and Role of Glial Cells in Herpes Simplex Virus- and Rabies Virus-Induced Encephalitis

**DOI:** 10.3390/v13122364

**Published:** 2021-11-25

**Authors:** Lena Feige, Luca M. Zaeck, Julia Sehl-Ewert, Stefan Finke, Hervé Bourhy

**Affiliations:** 1Institut Pasteur, Université de Paris, Lyssavirus Epidemiology and Neuropathology, 28 Rue Du Docteur Roux, 75015 Paris, France; lena.feige@pasteur.fr; 2Institute of Molecular Virology and Cell Biology, Friedrich-Loeffler-Institut (FLI), Federal Institute of Animal Health, Südufer 10, 17493 Greifswald-Insel Riems, Germany; Luca.Zaeck@fli.de (L.M.Z.); Stefan.Finke@fli.de (S.F.); 3Department of Experimental Animal Facilities and Biorisk Management, Friedrich-Loeffler-Institut (FLI), Federal Institute of Animal Health, Südufer 10, 17493 Greifswald-Insel Riems, Germany; julia.sehl-ewert@fli.de

**Keywords:** astrocytes, microglia, viral encephalomyelitis, herpes simplex virus, rabies virus

## Abstract

The environment of the central nervous system (CNS) represents a double-edged sword in the context of viral infections. On the one hand, the infectious route for viral pathogens is restricted via neuroprotective barriers; on the other hand, viruses benefit from the immunologically quiescent neural environment after CNS entry. Both the herpes simplex virus (HSV) and the rabies virus (RABV) bypass the neuroprotective blood–brain barrier (BBB) and successfully enter the CNS parenchyma via nerve endings. Despite the differences in the molecular nature of both viruses, each virus uses retrograde transport along peripheral nerves to reach the human CNS. Once inside the CNS parenchyma, HSV infection results in severe acute inflammation, necrosis, and hemorrhaging, while RABV preserves the intact neuronal network by inhibiting apoptosis and limiting inflammation. During RABV neuroinvasion, surveilling glial cells fail to generate a sufficient type I interferon (IFN) response, enabling RABV to replicate undetected, ultimately leading to its fatal outcome. To date, we do not fully understand the molecular mechanisms underlying the activation or suppression of the host inflammatory responses of surveilling glial cells, which present important pathways shaping viral pathogenesis and clinical outcome in viral encephalitis. Here, we compare the innate immune responses of glial cells in RABV- and HSV-infected CNS, highlighting different viral strategies of neuroprotection or Neuroinflamm. in the context of viral encephalitis.

## 1. Viral Infection Routes to Enter the Central Nervous System

### 1.1. General Considerations 

In contrast to other tissues, the micromilieu of the central nervous system (CNS) tightly controls brain immunity by different and unique immunomodulatory properties: the presence of tightly controlled physical barriers, the lack of a classical lymphatic drainage system, the presence of functional lymphatic vessels [1], the relative deficiency of constitutive MHC expression, and the immunomodulatory properties regulated via the expression of immunosuppressive proteins [2]. In the context of infection, access of pathogens to the CNS is limited by the blood–brain barrier (BBB), comprising a tightly connected endothelium and the glia limitans lining the CNS parenchyma [3]. If a pathogen gains access to the CNS, resident and innate immune cells mount innate immune responses, which lead to the recruitment of adaptive immune cells from the periphery. Under physiological conditions, immune responses in the CNS are limited to protect vulnerable and nonrenewable cells, such as neurons. Blood-borne viral pathogens, however, have evolved three different mechanisms to cross the neuroprotective BBB (1): (i) transcellular entry (West Nile virus, Zika virus, Japanese encephalitis virus), (ii) paracellular entry (Marburg virus, dengue virus, Ebola virus), and (iii) the so-called ‘Trojan horse entry’ in which the virus enters the CNS via circulating immune cells (HIV, Japanese encephalitis virus, measles virus, canine distemper virus) [4]. In contrast to paracellular transport, transcellular transport usually does not depend on the disruption of junction proteins, the actin cytoskeleton, or the basal lamina [4]. It is further subdivided into (i) free and passive diffusion limited to lipophilic substances, (ii) carrier-mediated transport, and (iii) vesicle-mediated transport [5]. In contrast, direct entry mechanisms (RABV, HSV, poliovirus, henipavirus) aim to bypass the neuroprotective barriers and enter the CNS via infection of nerve endings and long-distance transport of viral particles within the neuronal network (Figure 1) [2]. The exact mechanism of how viral pathogens such as RABV and HSV, both causing lethal encephalitis in humans, efficiently infect the CNS and how host immune responses contribute to or counteract efficient neuroinvasion remains a question of significant interest. 

Rabies, one of the oldest and most fatal diseases described in humans [8], is still responsible for approximately 60,000 human deaths per year [9]. About 50% of fatal cases occur in children, causing about 3.7 million disability-adjusted life years [10]. The causative agent of rabies is the rabies virus (RABV), a bullet-shaped negative-sense single-stranded RNA rhabdovirus belonging to the order *Mononegavirales*. RABV is mostly (>99%) transmitted to humans upon the bite of a rabid dog [6]. Thereafter, canine RABV variants replicate locally in striated muscle cells and spread to the peripheral nervous system (PNS) via the neuromuscular junction [11,12]. Compared with coyote street RABV (COSR), the silver-haired bat variant of RABV (SHBRV) has shown a more effective local replication in epithelial cells and fibroblasts, two cellular components of the dermis [13]. Thus, differences in RABV strains seem to define RABV tropism and replication potential at the entry site. Once inside the PNS, RABV is transported by motor neurons in the retrograde direction [14]. Dependent on the site of infection, it may also use sensory and autonomic neurons to reach the CNS [15]. Inoculation of the planar footpad of adult mice with RABV leads to the capture of viral particles by dorsal root ganglia and their transport to the spinal cord and later the brain [16]. Unlike necrotizing HSV encephalitis (HSE), RABV preserves an intact neuronal network until the final stage of the disease by inhibiting apoptosis and limiting inflammation within the CNS, two crucial factors ensuring RABV transmission [17]. 

Around a week after the onset of symptoms, major histopathological lesions are still absent in the RABV-infected CNS. Clinical onset of the disease develops after RABV is widely disseminated in the CNS. Neurons, which have been infected with RABV for several days, do not exhibit cytopathic changes and remain metabolically viable in vivo [18,19,20,21,22]. In mice, the development of severe clinical disease occurs along with protein expression changes in ion homeostasis as well as docking and fusion of synaptic vesicles to the presynaptic membranes [23], which is hypothesized to be responsible for the defective neurotransmission recognized in rabies [24]. In humans, rabies manifests in two clinical forms: furious rabies and paralytic rabies [25]. In both forms, RABV-induced dysfunction of the local dorsal root ganglia causes pain, paresthesia, or pruritus. After the prodrome, encephalitic patients show hypersalivation and periods of agitation alternating with lucidity, hydrophobia, and difficulty swallowing, ultimately leading to death. The paralytic form leads to an early onset of muscle weakness, followed by a longer time between disease onset and death or coma, compared with the encephalitic form. Following the onset of symptoms, rabies is almost invariably fatal [26]. In contrast to humans, several RABV-exposed mammal species produce RABV-neutralizing antibodies that clear RABV prior to CNS invasion [27,28,29]. As a result, subclinical or mitigated infections show mild or no symptoms. These nonlethal RABV infections have been shown or suggested for bats [27,28,30,31], cattle [29], wild canids [32,33,34], and nonhuman primates [34,35,36]. Although pre- and postexposure vaccinations against rabies are available [37,38], no treatment exists thus far, which shows a high therapeutic efficiency. In 2004, a young and unvaccinated patient from Wisconsin survived rabies [39]. The therapeutic approach used, more specifically the induction of a therapeutic coma and the use of N-methyl D-aspartate receptor antagonist therapy, is since then known as the Milwaukee protocol [39]. Despite its relentless promotion, there are at least 31 documented failures using the Milwaukee protocol, raising concerns about its efficiency [40]. Therefore, improving our understanding of the interplay between RABV and the host immune system may result in improved medical management of rabid patients.

Like RABV, the herpes simplex virus (HSV) invades the CNS by bypassing the BBB via retrograde transport along nerves. HSV-1 is a double-stranded DNA virus belonging to the highly neurotropic alphaherpesviruses. It is assumed that after initial replication in epithelial cells, HSV enters sensory nerve endings and establishes lifelong latency in peripheral sensory ganglia, such as the TG. Unlike RABV, alphaherpesviruses rarely spread to the CNS of immunocompetent hosts. Nevertheless, HSE does not occur more frequently in immunocompromised than in immunocompetent hosts, since it is believed that both virus-mediated and indirect immune-mediated mechanisms contribute to the observed CNS damages [41]. In contrast to immunocompetent patients, immunosuppressed patients might more likely show atypical or aggressive infections [42]. In sporadic cases, spread to the CNS can result from primary herpetic infection, reinfection by a second herpesvirus, or reactivation of latent HSV [43]. After CNS invasion, HSV-1 causes necrotizing meningoencephalitis [44]. In this regard, HSV-1 clearly differs from RABV since HSV-1 infection leads to a fulminant inflammatory response while this does not occur upon RABV infection. HSE, which is the most common form of sporadically occurring fatal encephalitis in humans worldwide, is mainly caused by HSV-1, with HSV-2 representing less than 10% of the cases, which are mostly affecting neonates. The neonatal form of HSE involves the periventricular white matter, mainly spares temporal and frontal lobes [45], and is rarely hemorrhagic [45,46]. In contrast, reactivation of HSV in the trigeminal sensory ganglion of older children and adolescents can lead to virus dissemination in temporal and frontal lobes [47]. 

To date, the mechanisms leading to CNS access and strong temporofrontal lobe tropism of HSV-1 are not fully understood. Unlike rabies, HSE caused by HSV-1 shows a bimodal age distribution and occurs in either children or adolescents younger than 20 years, or in adults over 50 years [48]. Although the pathogenesis of neonatal HSE is not well understood, it has been hypothesized that different routes of virus entry to the brain may account for different localizations of virus dissemination observed in HSE. In infants, HSV may spread hematogenously across the immature BBB [45]. In adults, however, HSV-1 most likely enters the CNS via the trigeminal or olfactory route. After nonspecific clinical signs, HSE patients develop neurological deficits as well as characteristic behavioral and personality changes, such as disorientation, speech disturbances, and focal or diffuse neurological signs, such as seizures or hemiparesis [49,50]. A prospective clinical study in 2007 showed that the high mortality rate of 70% of HSE can be reduced to 5% using acyclovir therapy [51]. However, survivors suffer from severe neuropsychological long-term sequelae, including mainly cognitive dysfunctions [52].

### 1.2. Which Factors Support RABV and HSV Replication in the Nervous System? 

RABV enters the cell via the interaction of the RABV glycoprotein (G-protein) with RABV entry receptors, triggering clathrin-mediated endocytosis of RABV particles [53,54,55]. One reason for a preferential infection of neuronal tissues might be the higher expression levels of RABV entry receptors on neurons and postsynaptic membranes of muscle cells in the neuromuscular junction [56]. Although the presence of RABV receptors might not be sufficient for the productive infection of non-neuronal cells due to inhibition by respective antiviral responses of the target cells, the presence of a specific receptor represents a prerequisite of infection. After viral entry into the cell, the RABV G-protein induces fusion of the viral envelope with the endosomal membranes liberating the viral nucleocapsid into the cytoplasm. In detail, the viral G-protein is a class III membrane fusion protein that undergoes a conformational change in the acidic environment of the endosome [54,57,58]. Like HSV, after entry at presynaptic axon membranes, RABV particles are transported in a retrograde fashion along microtubules [59,60,61]. Although the viral phosphoprotein (P-protein) has been shown to interact with a light chain of the dynein motor complex [62,63], axonal transport of RABV to the CNS is independent of this interaction [64]. More specifically, the binding domain of the viral P-protein seems to rather have a crucial role in RABV primary transcription compared with that in the intracellular transport of RABV [65]. For long-distance axonal transport, enveloped RABV virions are transported within internalized endosomes [60,61]. Upon arrival in the neuronal cell body, the viral polymerase L initiates transcription by using the ribonucleoprotein complex as the template [65]. Once a certain threshold of viral nucleoprotein (N-protein) is synthesized, viral replication is initiated, leading to the production of a full-length positive-sense RNA antigenome [66,67,68]. This intermediate serves as the template for the full-length negative-sense genome. The G-protein is inserted into the plasma membrane [69,70]. The connection of the viral ribonucleoprotein with the G-protein comprising membranes triggers membrane envelopment. Subsequently, the matrix protein (M-protein) mediates RABV budding [71] for transmission to next-order neurons. It is thought that RABV buds from the synapse to start another round of infection in next-order neurons [14]. Consequently, RABV uses trans-synaptic spread to reach the CNS. The brain area in which the infection starts depends on the innervation site of the infected motor neurons. From the brain, RABV centrifugally spreads to salivary glands and several extraneural organs [6,72,73]. In contrast to HSV [44], RABV does not persist in the nervous system and ultimately leads to the death of the host [26]. 

Unlike spreading via the neuromuscular junction, HSV-1 must overcome the barrier between epithelial cells and free-ending nerve fibers from the PNS. After initial replication, neuroinvasion of HSV-1 is mediated via the herpesviral conserved deubiquitinase domain of the tegument protein pUL36 [74,75]. Sensory neurons are readily infected, but if deeper layers of the epithelium are affected, also efferent and autonomic nerve fibers can be infected. Following fusion-mediated entry in neurons at axonal nerve endings, virus capsids and their inner tegument layer are released into the cell [76,77], which is in contrast to the clathrin-mediated and receptor-dependent endocytosis of RABV [55,78]. Comparable to intraendosomal RABV particles, HSV-1 nucleocapsids are then retrogradely transported along microtubules, which are mainly mediated by the viral proteins pUL36 and pUL37 [77,79] together with the cellular dynein motor complex [80]. Once in the cell body, virions dock at nuclear pores and deliver viral DNA into the neuronal nucleus [81]. Following primary infection, latency is established in infected neurons, where progression of productive infection and subsequent cell death are blocked [82]. During latency, the production of viral proteins is abrogated due to the inactivity of the viral transcriptional activator UL48; instead, the latency-associated transcript (LAT), a nonprotein coding RNA, is synthesized. LAT inhibits apoptosis of the infected cell and mediates the maintenance of the latent state [83]. After reactivation in sensory ganglia, newly formed viral particles are anterogradely transported back to the periphery, where initial infection originally started. Alternatively, the virus spreads to higher-order neurons via anterograde microtubule-based transport using cellular kinesin motor proteins. Transport back to the periphery leads to the development of HSV lesions (herpes labialis or cold sores), whereas invasion of second-order neurons and transport to the CNS is associated with severe disease [7]. 

## 2. The Role of Glial Cells in Maintaining CNS Homeostasis 

Inside the CNS, the neuronal network is protected from pathogens via resident glial cells, which can be divided into oligodendrocytes, astrocytes, and microglia. Oligodendrocytes are myelin-producing cells of the brain, which insulate neuronal axons and thereby facilitate rapid propagation of axon potentials [84,85]. Schwann cells (SCs) represent the counterparts of oligodendrocytes in the PNS and are responsible for forming myelin sheaths around peripheral neurons. In comparison, satellite cells, which are closely related to SCs, ensheath the cell bodies of sensory neurons located in trigeminal and dorsal root ganglia and share similarities with astrocytes [86,87]. Astrocytes are the most abundant cell type in the CNS and play an essential role in the initiation of viral defense in a type I interferon (IFN)-dependent mechanism. They monitor brain development and function, especially by controlling potassium levels, removing toxic substances, and modulating synaptic activity [88]. In contrast, microglia represent a rather small and unique myeloid cell population, which derive from primitive myeloid progenitors from the yolk sac [89]. Due to their similarity to macrophages, microglia are often referred to as the CNS-resident macrophages, which largely contribute to immunosurveillance in the brain [90,91]. Under physiological conditions, the homeostatic brain microenvironment constitutively expresses ‘off’ signals, which prevent astrocytic and microglial activation. Those microglia–astrocyte–neuron interactions are critical for CNS innate immunity [92]. In detail, neurons constitutively express surface molecules or secrete molecules, which inhibit microglial activity. For example, healthy neurons express CD200 to keep CD200R^+^ microglia in a quiescent state [93,94,95,96]. Additionally, glial-mediated production of IL-10 and related cytokines (IL-19, IL-24) limit glial inflammatory responses and promote immunosuppression in the CNS [97].

In the case of neuronal death or injury, the expression of those inhibitory molecules is modulated, leading to the abrogation of microglial and astrocytic suppression [93,94,95,96]. Damaged or stressed neurons activate astrocytes and microglia via the release of cytokines (IFN-γ, IL-6) and chemokines (CX3CL1, CCL2). In response, glial cells sense neuronal damage via cell-specific receptors and produce an array of cytokines (IL-1, IL-6, IL-12, TNF, CCL2, CCL4, CCL5, CCL7, CXCL10) depending on the specific pathogen [98]. This results in the upregulation of MHC I and II proteins on the microglial cell surface [99,100] and the increased expression of adhesion molecules on endothelial cells (ECs) [101,102]. Further, reactive astrocytes also present antigens via MHC I and II proteins during viral encephalitis and neuroinflammatory reactions [103,104,105]. Apart from neurons, numerous other players, such as peripheral immune cells and peripherally produced molecules, can induce glial activation and subsequent antigen presentation [106].

Infecting mice with a wild-type (wt) bat RABV has revealed the upregulation of microglial-specific CD200R in the virus-infected CNS, suggesting that RABV influences neuron–microglia signaling [107]. In HSE, microglia have been shown to sense HSV-1 through the cyclic guanosine monophosphate–adenosine monophosphate (cGAMP) synthase/stimulator of the interferon gene (cGAS/STING) pathway, initiating the production of type I IFN and immune priming of neighboring cells [108]. Apart from microglia, astrocytes are the main producers of IFN-β in the RABV-infected brain [109]. Similarly, astrocytic secretion of IFN-β has been shown in other viral CNS infections (e.g., with vesicular stomatitis virus (VSV) and La Crosse virus), suggesting that astrocytes are important immune modulators triggering viral clearance in the infected CNS [110,111]. 

Since morphological changes in the CNS of rabies patients [112] and infected animals appear mild despite the fatal outcome [113,114], it is believed that neuronal dysfunction is responsible for RABV-induced death [24]. The ability of RABV to antagonize an early IFN response enables the virus to limit apoptosis and inflammation in the CNS [115,116,117]. Consequently, the innate immune system fails to prime an adaptive immune response in time to clear the virus from the infected CNS. Unlike rabies, HSV infection initiates a fulminant inflammatory cascade, recruiting innate immune cells and activating adaptive immune mechanisms to the infected CNS area (Figure 2).

Although we are far from understanding the exact molecular pathways underlying neuron–glia interactions in the virus-infected CNS, RABV and HSV both modulate the gene expression of signaling molecules, which are important for neuron–glia communication, thereby interfering with CNS homeostasis. Taken together, the outcome of viral infection depends on two main factors: the pathogenicity of the virus and the triggered antiviral response. The latter is mainly mediated by glial cells in the CNS. Both viral pathogenicity and host immune response modulate one another, and their synergistic effects can result in adverse outcomes, due to either a highly pathogenic virus or an inadequate immune response. At the end stage of the disease, RABV leads to an invariably fatal encephalomyelitis by evading the immune response of glial cells, subsequently enabling RABV to extensively replicate in the CNS. In contrast, HSV mediates death by inducing necrotizing inflammation in the CNS, a process that is aggravated by an excessive immune response. 

## 3. The Cellular Infection Pattern of RABV and HSV in the Nervous System

As elaborated earlier, RABV and HSV are classically referred to as neurotropic viruses since both preferentially spread to the nervous system [14,118]. Whereas most of the research has succeeded in linking immunoevasive mechanisms of RABV to distinct viral proteins [119,120], it is less understood how they influence the strong neuron-specific tropism of RABV or why in some cases glial cell infection can be detected [121,122,123,124,125]. Deciphering the requirements for a strict neuron-specific tropism or broader host cell multitropism in the nervous system and discovering cell-specific responses restricting viral replication is of utmost importance to understand viral pathogenesis and antiviral host mechanisms. 

### 3.1. Viral Entry 

Regarding viral tropism, a broad range of cell types, including neuroglia, can be infected in vitro by both field and laboratory-attenuated RABV [122,126,127,128], contrasting the pronounced neuron-specific tropism in vivo. It is suggested that the neuron-specific tropism of RABV presents a viral adaptation mechanism to avoid immune detection by the host [129]. Different RABV receptors bind to the viral G-protein, of which the neuronal cell adhesion molecule (NCAM) [130], low-affinity nerve growth factor receptor (p75NTR) [131], or metabotropic glutamate receptor subtype 2 (mGluR2) [132] presents bona fide neuronal surface receptor molecules, whereas the nicotinic acetylcholine receptor (nAChR) [133] is expressed in muscle cells and may support RABV amplification prior to neuron infection through motoneuronal endplates. However, RABV receptor expression in mice and humans [134,135] does not positively correlate with cellular susceptibility to RABV infection, implying that receptor usage alone is not determining the strong neuron-specific tropism of RABV. Recent single-cell sequencing data of human brains revealed that the expression of p75NTR, mGluR2, and nAChR is relatively low in the human brain, whereas NCAM is highly expressed. More specifically, NCAM has been predominantly found on neuronal cells but shown to be also highly expressed on oligodendrocytes and certain astrocyte subsets. In contrast, microglia have shown low or no expression of NCAM in the human brain [136].

A major factor determining tissue- and cell-specific permissivity to HSV infection is the cellular receptor mediating entry of the virus. For HSV-1 cellular entry, different glycoproteins (glycoprotein (g)D, gB, gH, and gL) are required to bind to specific cellular receptors. Initially, gB and gH attach to cell-surface heparan sulfate proteoglycans, which are present in various cells, including neurons. Next, gD binds to the cell surface receptors nectin-1 (also referred to as poliovirus receptor-like protein 1) or TNF receptor superfamily member 14 (TNFRSF14), also known as herpesvirus entry mediator (HVEM), which mediates the activation of the gH/gL complex and subsequently the transformation of gB into a fusogenic state [137]. For successful fusion of the viral envelope with the cell membrane, a gB receptor such as paired immunoglobulin-like type 2 receptor α (PILRA), another paired-type receptor with homology to PILRA, myelin-associated glycoprotein, and myosin heavy chain 9 (MYH9, also known as NMMHC-IIA) is required [137]. The gD receptor nectin-1 is a member of the immunoglobulin superfamily and is located at cellular adherens and tight junctions involved in the cell–cell adhesion of epithelial and endothelial cells [138]. Most likely, nectin-1 presents the most important receptor for HSV-1 infection of epithelial and neuronal cells [137,139]. Although HVEM is widely expressed in the liver, kidney, and lung, as well as in T and B lymphocytes, leukocytes, epithelial cells, fibroblasts, and neurons on which it controls diverse proinflammatory and inhibitory signaling pathways [140,141], it seems to be largely irrelevant for the course of HSE since the patterns of HSV replication in the brain of wt and HVEM knockout mice have been indistinguishable [139]. The distribution of HSV-1 receptors has been examined in the human [141] and murine brain [142]. Specifically, nectin-1, TNFRSF14, and MYH9 have been highly expressed in the human hippocampus, and it has been shown for nectin-1 in the murine hippocampus to be the underlying cause for the higher susceptibility of this brain area to HSV-1 infection. Abundant nectin-1 levels have also been found in sensory, sympathetic, and parasympathetic nerves [143]. Immunohistochemical studies on nectin-1 expression in neuronal and non-neuronal cells of the human nervous tissue have revealed that neurons of the basal ganglia, cerebral cortex, diencephalon, brainstem, spinal cord, and peripheral dorsal root and trigeminal ganglia are highly positive. Further, ependymal cells, choroid plexus epithelial cells, and vascular endothelium have shown strong nectin-1 expression. Variable but still positive immunoreactivity has been present in oligodendrocytes, astrocytes, pericytes, SCs, and satellite cells, demonstrating potential target cells for HSV-1 [142].

In a nutshell, multiple receptors are known to be responsible for the cellular uptake of RABV and HSV. Like RABV, not all the cells expressing the viral entry receptors get predominantly and productively infected by HSV, suggesting that more factors shape HSV tropism.

### 3.2. RABV Tropism

Considering the remarkable tropism of field RABV strains for non-neuronal cells in comparison with less virulent laboratory strains [122] as well as the broad distribution of the potential RABV receptors in the central and peripheral nervous systems [56,136], evidence is accumulating that RABV neuron-specific tropism may not exclusively be a result of a specific receptor expression by neurons. More likely, a lower protective immune response in neurons as compared with other susceptible cell types may result in productive infection. In contrast, entry in non-neuronal cell types seems to result in abortive infection due to immediate and strong innate immune responses [109,144] in a virus-strain-dependent manner [122]. Considering reports about in vivo infection of non-neuronal cells by field RABV [122,123,145,146,147], the concept of an almost exclusive infection of neurons has to be relativized. In summary, RABV host cell tropism and in vivo spread is due to multiple factors: the presence of RABV cell entry receptors on the plasma membrane, intraneuronal long-distance virus transport processes both at virus entry and after replication, specific trans-synaptic spread between neurons, and the neuronal metabolism, which actively supports RABV replication. Paracrine effects through cytokine release by abortively or productively infected non-neuronal cells such as astrocytes or SCs surrounding neurons [122,123] may also determine the susceptibility and survival of neurons and other cell types that can be easily infected in vitro [127]. 

#### 3.2.1. Astrocytes

Besides receptor usage, negative regulation by cell-type-specific inhibition of virus replication in non-neuronal cells might play a decisive role in viral tropism of neurotropic pathogens. In vivo, abortive infection of astrocytes with RABV SAD carrying the G-protein of the more neurotropic challenge virus standard (CVS) strain has been suggested to be the result of type I IFN induction [109]. In vitro, this correlates with the activation of double-stranded (ds) RNA-dependent mitochondrial antiviral signaling protein (MAVS), whereas wt RABV is able to infect astrocytes without exhibiting MAVS activation [144]. Other studies using in vitro neuron or astrocyte cultures have confirmed astrocyte infection by RABV in vitro, without remarking discernible differences between lab-attenuated and field RABV [122,127,148]. A broad quantitative comparison of different recombinant field and lab-attenuated strains in infected mice, however, has highlighted the infection of astrocytes in the CNS after intramuscular infection only with highly virulent field RABV. In addition, no astrocyte infection has been detected after infection with lab-attenuated or vaccine strains [122]. Abundant expression of the RABV P-protein, a major IFN antagonist of RABV in astrocytes [115,149,150,151,152,153,154], suggests that inhibition or delay of innate immune responses in astrocytes may be a hallmark of pathogenic field RABV pathogenesis. Accordingly, nonabortive infection of astrocytes by pathogenic field RABV may critically influence the CNS microenvironment, including, for example, a failure to permeabilize the BBB upon infection, which has been linked to contribute to lethal disease progression. In summary, RABV infection of immune and neuron-regulatory astroglia strongly depends on the viral strain, and RABV pathogenesis may depend on the ability to abortively or productively infect these cell types in the brain. 

#### 3.2.2. Microglia

While activation of microglia has been observed during RABV infection in vitro and in vivo [155,156,157,158,159], the detection of viral antigen in infected microglia in vivo has been assumed to be the result of microglial phagocytosis of infected neurons [128]. Ray and colleagues provided some evidence that human microglia are susceptible to RABV infection in vitro [127]. Recently, the susceptibility of human induced pluripotent stem cell-derived microglia-like cells in vitro has been questioned [160]. Thus, more research is necessary to assess whether microglia productively support RABV infection. 

#### 3.2.3. Oligodendrocytes 

Like microglia, oligodendrocytes represent another comparatively small population of glial cells [161]. In contrast, oligodendrocytes are highly vulnerable to cytotoxic by-products, such as reactive oxygen species (ROS), high iron content, high glutamate concentrations, and high ATP concentrations [85]. However, little is known about the role of oligodendrocytes during RABV infection. Recent advances with monosynaptic viral RABV tracers have demonstrated occasional oligodendrocyte infection [161,162]. 

#### 3.2.4. Schwann Cells 

Whereas awareness about the role of CNS neuroglia in RABV pathogenesis has increased in the last years, a contribution of peripheral neuroglia such as SCs may have not been considered adequately. SCs are highly immunocompetent and exert key functions in immune regulation by antigen presentation, pathogen detection, and cytokine production [163,164]. Although release and virion accumulation between axonal and SC membranes as well as nodes of Ranvier has been demonstrated by electron microscopy, until recently, SCs were considered unsusceptible to lyssavirus infection [165,166]. A recent three-dimensional immunofluorescence imaging approach for high-volume tissue imaging [167], however, has enabled the detection of RABV-infected SCs in peripheral nerves after field RABV infection of mice via the natural (intramuscular) and an artificial (intracranial) inoculation route. Accordingly, a model has been suggested in which the infection of peripheral SCs suppresses glia-mediated innate immunity and delays antiviral responses to field virus infections [123].

### 3.3. HSV Tropism

Herpesviruses are pantropic, causing lytic infection in various tissues and cell types in vitro. In humans, *herpes labialis* is caused by HSV-1 infecting epithelial cells of the orolabial mucosa, followed by virus uptake and transport from closely adjacent axons of sensory nerve endings to ganglia of the PNS. Further, HSV-1 is able to infect the corneal or genital mucosal epithelium, causing herpes genitalis or herpes keratitis, respectively [168]. Once in the PNS, HSV-1 establishes lifelong latency within the sensory and autonomic peripheral ganglia of the head and spinal cord. It can be periodically reactivated by various stimuli [169,170]. 

HSV-1 tropism to the frontal and temporal lobe has been discussed given the respective invasion route and synaptic connections by which the virus may reach the brain, the olfactory and trigeminal nerve [171]. Thus, direct connections from the PNS to the CNS facilitate invasion towards the brain across synapse-linked neurons. However, detailed research is still needed on that point, especially on why HSV-1 maintains the devastating potential to reach the CNS in rare cases, but otherwise resides latently in peripheral ganglia [172]. Moreover, it is not yet clear whether HSV-1 reaches the CNS directly after primary infection, which leads to encephalitis, or whether this happens after reactivation of the virus in latent infected tissue [44]. This is in contrast to rabies, which is always virulent and invariably fatal when entering the nervous system [50].

#### 3.3.1. Astrocytes 

Apart from neurons, glial cells have been shown to be permissive to HSV-1 infection [173,174]. Lokensgard and colleagues reported that astrocytes are highly susceptible to productive HSV-1 infection and fail to produce cytokines upon infection [173,174]. Other studies, however, have reported the upregulation of pathogen recognition receptors, DNA/RNA sensors, IFNs, and IFN-stimulated genes (ISGs) in astrocytes upon HSV-1 infection [175,176,177]. Upon in vitro and in vivo infection, astrocytes respond to infection with hypertrophy and rounding and syncytia formation [178,179,180]. 

#### 3.3.2. Microglia

In microglia, HSV-1 infection does not result in productive viral replication, but instead in the production of significant amounts of proinflammatory cytokines [181]. In HSV-1 infected brains, microglia are present around cell bodies and dendrites of neurons before mature viral particles are detectable [182]. 

#### 3.3.3. Oligodendrocytes

Apart from astrocytes, human oligodendrocytes have been shown to be variably permissive to HSV-1 infection in vitro [183,184]. However, in HSV-1 infected mice, HSV-1 antigen-positive oligodendrocytes have been found in the spinal trigeminal tract [185]. In line with this, the occurrence of demyelinating lesions in rodents supports the susceptibility of oligodendrocytes to HSV-1 infection [186]. 

#### 3.3.4. Schwann Cells 

It has been shown that murine SCs can be infected after intraocular inoculation, but produce no viral progeny, whereas direct inoculation into the sciatic nerve leads to productive HSV-1 infection [187]. In another study, infection of SCs was demonstrated only at late time points after infection, confirming the notion that these cells (i) are not the primary target of HSV-1 and (ii) do not play a role in retrograde transport upon neuroinvasion. Thus, it is very likely that SCs become infected via productive infection of the neuronal cell body, the subsequent escape of HSV-1 along axons, and the emergence into associated SCs along the nerve tract [188]. Similar to SCs, the infection of satellite cells seems to be abortive since only empty capsids have been detected by electron microscopy of chicken dorsal root ganglia [189]. In line with this, very limited infection has been reported in satellite cells from mouse and human dorsal root ganglia [190,191]. 

Taken together, RABV field strains and HSV-1 productively infect neurons and astrocytes in vivo, while microglia and oligodendroglia are less or not at all susceptible to infection. Infection of SCs has been observed for field RABV and HSV infection, but the role of infected SCs in viral encephalitis remains poorly understood. On the one hand, the strain specificity of the virus and, on the other hand, receptor expression and cell-specific immune responses of the cellular host determine cellular tropism and infection outcome. Although it is suggested that RABV and HSV bear a neuron-specific tropism, cellular tropism might be much wider than initially believed (Figure 3). 

## 4. The Role of the Blood–Brain Barrier in HSV and RABV Infections 

Blood vessels vascularizing the CNS deliver oxygen and nutrients to the nervous tissue, analogous to blood vessel functions in peripheral organs. Apart from this, however, the BBB possesses unique properties allowing the precise control of molecule and cell movements between the blood and the brain. Physiological barrier properties are mediated by ECs, which are tightly regulated by vascular, immune, and neural cells. The main function of the BBB is the protection of neural tissue from toxins and pathogens [3]. Conversely, alterations of BBB functions endanger CNS homeostasis and are implied in numerous neurological diseases, such cerebral ischemia, brain trauma, brain tumors, and brain infections [194]. 

### 4.1. Rabies Virus

Infection of nerve endings and retrograde axonal transport enables RABV to bypass the neuroprotective BBB, which normally restricts viral entry [195]. Subsequently, RABV benefits from the immunologically quiescent neural environment of the CNS [196]. More importantly, viral protein-mediated sequestration of the immune response, especially its antagonism against early IFN response, enables RABV to successfully replicate inside the CNS [119,197]. RABV-mediated sequestration of the host immune response hinders the opening of the BBB, leading to lethal RABV infection [198,199,200]. Accordingly, Roy and colleagues showed that mice infected with the lethal SHBRV develop a fully functional immune response in the periphery, while immune cells are unable to migrate to the infected CNS [201]. As a consequence of deficits in increasing the BBB permeability in the cerebellum, immune effector cells and associated virus-neutralizing antibodies cannot reach the CNS for RABV clearance [113,114,202]. However, the exact mechanisms linking RABV-mediated sequestration of the host immune response to the failure to open the BBB remain to be elucidated. 

Even though an effective therapy for rabies is still missing, studies have reported the survival of a few RABV patients with acute illness [203,204,205,206,207]. One of the main findings associated with nonlethal RABV infection is the presence of virus-neutralizing antibodies in the serum or CSF of RABV survivors [208,209]. In mice, crossing of virus-neutralizing antibodies through a permeable BBB protects against lethal RABV infection [199,210]. By comparing mice with different genetic backgrounds, Roy and Hooper showed that the opening of the BBB is of utmost importance to the clearance of RABV from the infected CNS [198]. Accordingly, SJL mice known to develop more extensive CNS inflammation due to the increased recruitment of infiltrating IL-10 producing CD4^+^ cells [211,212] more often survive wt bat RABV infection compared with other mouse strains [213,214]. In contrast, 129/SvEv mice succumb to infection with SHBRV due to the failure of opening the BBB [200]. To increase BBB permeability, SJL mice were immunized with myelin basic protein to induce experimental allergic encephalomyelitis prior to SHBRV infection. Immunized and infected mice showed higher survival rates compared with nonimmunized and infected mice. Vice versa, mice with reduced BBB permeability after treatment with the steroid hormone dehydroepiandrosterone presented higher morbidity, pointing out that the infiltration of immune effector cells across the BBB is critical to survive RABV infection [198]. Emphasizing the importance of immune-response-stimulating agents, intracerebral administration of recombinant RABV expressing the granulocyte-macrophage colony-stimulating factor (LBNSE-GM-CSF) protected more mice from developing rabies compared with the administration of UV-inactivated LBNSE-GM-CSF. LBNSE-GM-CSF infection led to significantly higher levels of chemokine and cytokine expression and more infiltrating immune cells in the CNS than UV-inactivated LBNSE-GM-CSF. Additionally, direct administration of immune-response-stimulating agents to the CNS enhanced BBB permeability and infiltration of immune cells, consequently preventing the development of rabies [198]. Overall, the failure of opening the BBB during RABV infection enables the virus to successfully replicate in the tightly controlled immunosuppressive micromilieu of the CNS. Clearance of RABV from the CNS requires RABV-specific immunity, such as the presence of circulating virus-neutralizing antibodies and the enhancement of BBB permeability, to allow immune effectors to access the CNS. Concomitant peripheral and intracerebroventricular administration of the two potent neutralizing monoclonal antibodies, RVC20 and RVC58, directed against the viral G-protein cured symptomatic rabid mice [215,216]. Antibody-based therapeutic approaches remain a promising treatment to treat neurological symptoms and cure symptomatic rabies. 

### 4.2. Herpes Simplex Virus

In contrast to rabies, the disruption of the BBB is a crucial pathological mechanism in the development of HSE, which has been recently reviewed in detail [217]. Brain damage is mediated by lytic infection of neurons and glial cells as well as by neuroinflammatory processes [218]. Toll-like receptor (TLR)-dependent recognition of HSV-1 by neurons and glia leads to the initial production of cytokines, such as type I IFN, IL-6, IL-1β, IFN-γ, and TNF, resulting in the disruption of the BBB [219]. Consequently, immune cells are recruited to the brain parenchyma, resulting in an enhanced inflammatory response to HSV-1 infection and increased neuronal damage. In vivo, CD4^+^ or CD8^+^ T cells and macrophages are present in perivascular infiltrates close to HSV-infected cells in the murine brainstem [220]. In addition, vasogenic edema and hemorrhage may occur [217]. In general, leukocyte entry into the CNS is regulated by several factors, including the expression of adhesion molecules (PECAM, CD99, VE-cadherin, and JAM-A, B, and C) [221], expression of matrix metalloproteinases (MMPs) [222,223], and endothelial expression of chemokines (CCL2, CCL5, CXCL10, CXCL8) [224]. Briefly, HSV-1 affects the structure and function of the BBB, likely via inflammatory-induced activation [225] and direct infection of ECs [226,227]. More specifically, HSV-1-induced increase in ICAM-1 expression on ECs mediates an increased interaction between ECs and circulating leukocytes and subsequently leads to infiltration of immune cells into the brain parenchyma [225,227]. Very recently, it could be demonstrated that HSV-1 alters the Golgi structure in ECs of the BBB and thus reduces the integrity of the same [228]. Further, MMPs that are thought to cleave tight junction proteins of the BBB are significantly upregulated in HSV-1-infected humans and mice. High levels of MMP2 and MMP9 have been shown to play a pivotal role in BBB damage in HSE [229] and other brain infections, such as West Nile virus encephalitis [230]. ROS, which are increasingly produced by microglia in HSE, are both beneficial and disadvantageous. On the one hand, they prevent the further spread of the pathogen; on the other hand, they degrade tight junction proteins and thus increase the permeability of the BBB [231]. 

However, how HSE-associated ROS alters tight junction integrity urges further research. CCL2, a major chemokine in CNS inflammation, also plays an essential role in the regulation of the BBB permeability, endothelial dysfunction, and increased leukocyte recruitment [232]. Different molecular pathways include the reorganization of the actin cytoskeleton of ECs mediated through the interaction between CCL2 and its receptor CCR2 [233] or the disruption of adherens junctions by Src-kinase-dependent phosphorylation of β-catenin [232]. Dysfunction of the BBB can also be caused by astrocyte disturbances. HSV-1 has been reported to increase the expression of aquaporin 4 (AQP4) on astrocytes, which regulates water transport across the BBB [234]. An increase in AQP4 leads to water influx into astrocytes and the basement membrane, indirectly leading to the disruption of the BBB [235]. In addition, this mechanism leads to intracellular or extracellular brain edema, which is associated with a high mortality in HSE patients [236]. Furthermore, HSV-1 infection of astrocytes leads to apoptosis through the interaction between the virus and mitochondria, resulting in the destruction of the BBB [217]. 

Whether reactive astrocytes are beneficial in HSE remains under debate. In an HSE mouse model for chronic inflammation, it was demonstrated that astrogliosis occurs acutely and chronically after infection and is associated with a persistent inflammatory, potentially harmful reaction [237]. In contrast, reactive astrocytes are able to release vascular protective factors, which promote the repair of the BBB, thereby decreasing the expression of endothelial surface receptors and leukocyte infiltration [238]. In summary, HSV-1 infection of the brain, in contrast to RABV infection, leads to compromised BBB function and subsequently to a fulminant immune response, whose neuroprotective versus neurodestructive properties have yet to be defined.

## 5. Differences in Cell-Type-Specific Innate Immune Responses between RABV and HSV Infection in the CNS

The innate immune system represents the first line of defense against viral invaders. Germline-encoded pattern recognition receptors (PRRs), such as TLRs and retinoic acid-inducible gene I (RIG-I)-like helicases (RLRs), sense evolutionary conserved pathogen-associated molecular patterns and danger-associated molecular patterns [239]. Further, attention has recently been drawn to cytosolic PRRs, which sense cytosolic DNA, such as cGAS, gamma-interferon-inducible protein (IFI16), and absent in melanoma 2 (AIM2) [240]. All major glial cells, more specifically astrocytes, microglia, oligodendrocytes, and SCs, as well as neurons, express a different repertoire of PRRs including TLRs. In detail, TLR2 and TLR9 are expressed by microglia and astrocytes, whereas TLR3 is found in microglia, astrocytes, oligodendrocytes, and neurons [175,241,242]. Further, RLR RIG-I and melanoma differentiation-associated protein 5 (MDA5) are present in microglia, astrocytes, and neurons [243]. Apart from differences observed in the expression of PRRs between the distinct CNS cell types, PRR expression varies tremendously due to its tight regulation by the cellular differentiation status, the inflammatory processes, and the presence of pathogens [244]. Generally, viral recognition via PRRs leads to an early IFN response, which restricts viral growth. Another essential function of the innate immune system is to prime and initiate an adaptive immune response [245]. Compared with glial cells, it is thought that neurons may lack robust innate immune signaling to avoid damage to this largely irreplaceable and nonrenewable cell population [246]. Overall, distinct cell-specific innate immune responses to viral infection may account for viral tropism and pathophysiological findings, such as a rather quiescent versus fulminant inflammatory response (Figure 4). 

### 5.1. Rabies Virus

RABV is immediately recognized by the innate immune system after the virus is exposed to the skin or muscle by bites or scratches. Innate PRRs sense RABV at first in the periphery and afterwards in the CNS [254]. Intracellularly, RABV is recognized via MDA5 and RIG-I [109,144,255], TLR3 [256,257,258], or TLR7 [259,260,261], consequently leading to the activation of the NF-κB (nuclear factor ‘kappa-light-chain-enhancer’ of activated B-cells) pathway and the secretion of type I IFNs [245]. 

After cellular recognition of RABV by the innate immune system, macrophages play crucial roles in the uptake of pathogens via specialized endocytic mechanisms [262]. Lytle and colleagues observed that after rabies immunization, macrophages accumulate at the entry site [263] probably due to the chemokine-mediated recruitment of circulating monocytes. Like macrophages, dendritic cells (DCs) also efficiently ingest circulating pathogens and display antigens to prime adaptive immune responses [262]. Infection with lab-attenuated RABV leads to the RIG-I-dependent activation of DCs and subsequent production of type I IFNs [255,264]. In contrast, binding and entry into DCs by wt RABV is severely blocked, and infection does not efficiently induce DC activation [265]. Despite the inability of DCs to activate T cells during wt RABV infection, activated CD69^+^ T cells can be found in the draining lymph nodes and peripheral blood of infected mice, independent of the virulence of the RABV strain [198,266].

Once the virus enters the CNS, RABV triggers innate immune responses of neurons [256], astrocytes [109,144], and microglia [109,249], which mount primary (activation of NF-κB) and secondary IFN responses (activation of Janus kinase–signal transducers and activators of transcription (JAK–STAT)), inducing the production of ISGs, such as cytokines and chemokines [245]. However, the extent of immune responses triggered strongly depends on the cell type [109] and the viral strain used [114,122,144,267]. Cytokines and chemokines are crucial to recruit peripheral immune cells to the CNS, since both favor changes in BBB permeability, which in turn regulates the contact between circulating immune cells and the neural tissue [101]. Since wt RABV stimulates little or no inflammatory response, recruitment of peripheral cells to the CNS is limited [200,201]. 

#### 5.1.1. Neurons

TLR3 is involved in RABV pathogenesis [257,268] and is expressed by glial cells and neurons [88]. Upon RABV infection, human neuronal NT2-N cells upregulate TLR3 expression in vitro after challenge with the RABV strain CVS, possibly due to sensing of viral dsRNA. TLR3, subcellularly located in endosomes, senses endocytosed dsRNA [269]. In a human neuroblastoma cell line, CVS infection leads to the localization of TLR3 in spherical perinuclear structures [257] called Negri bodies (NBs), which represent active RABV replication sites [270,271]. Together with enhanced TLR3 expression in cerebral cortical tissues in rabies patients [258], this leads to the hypothesis that RABV inactivates the physiological function of TLR3 as a viral sensor and exploits the protein to help the spatial arrangement of RABV-induced NBs to support viral replication [257,271]. Additionally, taking into account that TLR3 can mediate neuronal apoptosis [272], sequestration of TLR3 might allow RABV to escape TLR-mediated apoptosis. Further, Peng and colleagues provide evidence that wt RABV induces autophagy, which can in turn decrease apoptosis in neuroblastoma cells in vitro [247]. During RABV neuroinvasion, RABV limits apoptosis and inflammation to ensure the integrity of the neuronal network, subsequently enabling RABV transport throughout the CNS [117]. Following CVS infection, human neuronal NT2-N cells strongly upregulate the expression of IFN-β, CCL5, CXCL10, IL-6, TNF, and IL-1α [256]. There is increasing evidence stating that IFN-β [273], CCL5 [274], and IL-6 [275,276] also harbor neuroprotective functions. Despite the onset of rabies-induced neurological signs and symptoms, histological lesions stay relatively mild in the RABV-infected CNS [277]. Apoptosis has been reported in neuronal cells when infected by RABV, and several mechanisms have been investigated [278,279,280,281]. Although necrosis of several areas might appear sporadically in the brain of human rabid patients [112,282], neuronal necrosis in RABV-infected mice is highly restricted and depends on the viral strain used for infection experiments [283,284,285]. Apart from necrosis, apoptosis is strongly limited in RABV-infected murine, canine, and human brains [192]. Still, neurons exhibit marked beading and fragmentation of dendrites and axons, as well as swollen mitochondria in perikarya and proximal dendrites in the CVS-infected murine cerebral cortex [286]. Although we are far from fully understanding cellular death pathways involved in rabies pathogenesis, it is believed that both structural changes of neurons [286] and neuronal dysfunction [24] can explain the fatal outcome of rabies rather than neuronal cell death.

#### 5.1.2. Astrocytes

Dependent on the virus strain, astrocytes can support RABV infection in vitro and in vivo [122]. In general, astrocytes have higher expression levels of ISGs, rendering them less susceptible to viral infections than other cell types [287]. There is growing evidence that inflammatory upregulation of TLR3 expression in human astrocytes results in neuroprotection [288]. However, the role of TLR3-mediated signaling in astrocytes during RABV infection is still unknown. Besides TLRs, RLRs represent another group of PRRs, which sense RABV [109] and consequently trigger antiviral responses via the activation of NF-κB and JAK–STAT pathways [289]. The RLRs MDA5 and RIG-I are cytoplasmic viral sensors, which recognize viral replication via recognition of dsRNA, resulting in MAVS-mediated IRF3 activation [264]. In astrocytes, lab-attenuated RABV produces higher levels of dsRNA compared with wt RABV. As a result, lab-attenuated RABV induces MDA5/RIG-I-mediated activation of MAVS/p38/NF-κB to a higher degree than wt RABV. Consequent production of ISGs and inflammatory cytokines leads to the clearance of lab-attenuated RABV from astrocytes. In contrast, wt RABV nonabortively infects astrocytes through the evasion of the MDA5/RIG-I sensory pathway [144]. Primary murine astrocytes respond with the expression of IL-6 and TNF to VSV infection, a rhabdovirus that closely resembles RABV. In contrast, heat-inactivated VSV robustly lowers astrocyte-mediated cytokine expression, viral replication, and sensing of viral RNA by RLRs [290]. Overall, RLRs play a crucial role for RABV sensing and initiating cytokine production in infected astrocytes; however, more experimental data are needed to confirm this hypothesis. 

In vivo, astrocytes induce IFN-β production via RLR and TLR signal transduction pathways, representing the main source of IFN-β in the virus-infected brain [109]. For other neurotropic virus infections, astrocytes have also been implicated as the main type I IFN producers in the CNS [110,111]. Moreover, lab-attenuated RABV induces the expression of a variety of inflammatory cytokines (TNF, IL-6, IL-1β, IFN-γ, IL-17, and VEGF) in astrocytes, which in turn regulate BBB permeability [144]. Generally, astrocytes are also known as innate immune neuroglia releasing a variety of neurotrophins and pro- and anti-inflammatory mediators (e.g., TGF-β, IL-1β, IL-6, CXCL10, CXCL12), enabling the communication with neighboring cells and recruiting adaptive immune cells to the CNS [101]. Interestingly, it has been shown that pathogenic field RABV infection strongly modulates microRNA (miRNA) expression in the brains of infected mice. Predicted targets of RABV-modulated miRNAs involve the TGF-β signaling pathway [291], which is strongly associated with astrocyte scar formation [292]. Other targets of RABV-modulated miRNAs include the JAK–STAT signaling pathway (especially JAK2), suppressor of cytokine signaling (Socs3), and Socs4 [291]. It is well known, that the JAK2–STAT3 pathway plays a central role in astrocyte reactivity upon multiple pathological conditions [293], making them a possible target of RABV-modulated miRNAs.

#### 5.1.3. Microglia

Besides immunologically active astrocytes, microglia respond to infection via expression of ISGs, chemokines, and proinflammatory cytokines [263]. In detail, macrophage-mediated endocytosis of inactivated and infectious RABV virions activates the extracellular signal-regulated kinases 1/2 (ERK1/2) signaling pathways, ultimately leading to chemokine expression [294]. Chemokine secretion mediates leukocyte recruitment into the CNS in a multistep process [295]. Not only RNA levels of CXCL10 and CCL5 in the RABV-infected murine microglial cells Ra2 are approximately 3000-fold higher compared with mock-infected cells, but also protein levels increase steeply, reaching a plateau phase at 20 h postinfection. Furthermore, the expression of CCL5 and CXCL10 is modulated upon RABV replication since UV-inactivated virions do not trigger chemokine expression [249]. Emphasizing the importance of the mitogen-activated protein kinase (MAPK) subfamilies p38, c-Jun *N*-terminal kinases (JNK), and ERK 1/2 in the establishment of an antiviral response against RABV [294,296], Nakamichi and colleagues discovered that RABV triggers phosphorylation of MAPK signal transduction pathways in murine microglia, possibly triggering chemokine secretion [249]. Interestingly, activation of p38 and ERK1/2 signal transduction pathways stimulates CXCL10 expression, whereas CCL5 expression is positively regulated by p38 and negatively regulated by ERK1/2. Particularly p38 is required for the NF-κB-mediated expression of CXCL10 and CCL5 [249], chemokines known to act as proinflammatory mediators during viral encephalitis [297,298,299,300]. Apart from the secretion of cytokines and chemokines, microglia also produce important amounts of type I IFN during RABV infection [109].

### 5.2. Herpes Simplex Virus

HSV-1 infection is controlled by both the innate and adaptive immune system and is initially recognized by TLR2, TLR3, and TLR9 [301]. While HSV glycoproteins are sensed by TLR2 on the cell surface, viral DNA is detected by endosomal TLR9, followed by the activation of cGAS and the adaptor protein STING. Besides TLRs, RLRs, including RIG-I and MDA5, are described to detect HSV-1 RNA in the cytoplasm of infected cells, leading to the activation of the MAVS adaptor, NF-κB, activator protein 1 (AP-1), and IFN regulatory factors (IRFs) such as IRF3 and IRF7. These molecules act as transcription factors, which result in the production of proinflammatory cytokines and type I IFN [302], triggering the initial response to HSV-1 infection [303]. IFN-α and IFN-β signal through the interaction with interferon receptors (IFNAR) 1 and 2, resulting in the subsequent activation of the JAK–STAT signaling pathway [304]. More specifically, it has been shown that the DNA-dependent RNA polymerase III (Pol-III) is responsible for transcribing cytosolic viral DNA into an RNA ligand that binds to RIG-I, resulting in the induction of IFN-β upon HSV-1 infection [305]. In the CNS, HSV-1 is mainly recognized by microglia and astrocytes. 

The immune response in HSE has been described as a ‘double-edged sword’. It is crucial to control viral replication early after infection, as otherwise an uncontrolled immune response can lead to excessive reactions that can be fatal for the host [306]. The first-line viral defense is mediated by the TLR2 receptor driving the expression of various interleukins, type I IFN, and TNF [307]. However, since mice lacking TLR2 have revealed higher survival rates after HSV-1 infection than the wt strain, the TLR2-mediated response can have detrimental effects on the outcome of HSE [308]. In contrast to TLR2, TLR9 does not play a major role in the fight against HSV-1 [309], but rather functions simultaneously with TLR2 by recruiting natural killer (NK) cells to the CNS [310,311]. For the control of HSV-1 replication and the production of type I and II IFNs in the brain, TLR3-mediated sensing of HSV-1 has been described as extremely important [312]. Apart from TLR3, the cytosolic PRRs RIG-I and DNA-dependent activator of interferon regulatory factor (DAI) have been shown to restrict HSV-1 replication [313,314]. Unlike HSV-1, wt RABV successfully evades TLR3 signaling, thereby inhibiting the induction of IFN [257,271]. 

#### 5.2.1. Neurons

Neurons are highly susceptible to HSV infection. Lafaille and colleagues provide strong evidence that neuronal induction of IFN-β and/or IFN-γ is highly dependent on UNC-93B, which is associated with TLR-mediated responses [315]. Further, UNC-93B-deficient human neurons exhibit impaired IFN-β and IFN-γ responses to HSV-1 infection and are more susceptible to HSV-1 infection [315]. Although secretion of IFNs impairs HSV-1 replication in iPSC-derived human neurons [315], type I IFN treatment fails to completely block HSV-1 replication in murine neurons [246]. Similar to RABV, autophagy, not IFN signaling, is suggested to be the dominant antiviral strategy employed by neurons to control HSV infection [246,248]. Using an in vitro system of purified TG neurons from adult mice grown in compartmentalized chambers, Rosato and colleagues showed that the administration of IFN-β at either the soma or the axon is capable of restricting HSV-1 replication. Moreover, they showed that non-neuronal IFN signaling is insufficient to control HSV-1 dissemination and mortality in vivo, whereas neuronal IFN signaling alone is necessary to control HSV-1 replication, disease, and survival [316]. Recently, studies have highlighted the importance of the neuronal differentiation state and subtype on antiviral signaling capabilities [317,318,319]. Thus, different models and differentiation states might explain differences observed in the role of the neuronal type I IFN response. Taken together, neurons might restrict HSV replication via the induction of autophagy [246,248] and type I IFNs [315,316]. 

#### 5.2.2. Astrocytes

Astrocytes have not been intensively studied in HSV-1 brain infection. Recently, the expression of TLRs in response to HSV-1 infection in astrocytes has been investigated in more detail [175]. Mainly TLR2, TLR6, TLR7, TLR8, and TLR9, as well as DAI, cGAS, and MDA5, are upregulated by astrocytes, while IFI204, which senses nuclear DNA, is downregulated, implicating a potential viral immune evasion strategy. Upon infection, astrocytes react with the production of type I and II IFNs, which in turn activate the JAK–STAT pathway. Especially STAT4 expression is increased, which may stimulate the proinflammatory IL-12 cascade. In addition, different ISGs have been identified, which elicit antiviral properties in response to HSV-1 infection of astrocytes, such as RNase L, PKR (protein kinase R), IFIT (interferon induced proteins with tetratricopeptide repeats), and viperin. In contrast to microglia, astrocytes do not produce any neurotoxic substances [320]. Moreover, TLR3 expression in astrocytes is necessary to respond to neurotropic HSV-2 infection with the induction of IFN-β production possibly preventing viral spread [252].

#### 5.2.3. Microglia

Cytokines produced by glial cells play a crucial role in the activation of glia in the CNS itself and in the attraction of immune cells, such as DCs, NK cells, and lymphocytes, to the brain [253]. Specifically, human microglia produce high amounts of CCL5, CXCL10, TNF, and IL-1β as well as lower amounts of IL-6, IL-8, CCL3, CCL4, and CCL2 in a TLR3-dependent manner [193]. Similar to astrocytes in RABV pathogenesis, microglia are the main source of type I IFN in HSE [108]. Those glial cell-produced factors reveal variable antiviral properties in vitro. Whereas TNF drastically limits viral replication in astrocytes, IL-1β decreases viral replication comparatively little. In neurons, however, TNF and IL-1β have no effect on viral replication, whereas CXCL10 drastically decreases HSV-1 replication [193]. For HSV-1, it could be shown that microglia transfer its antiviral response to neurons and astrocytes [108]. Through the expression of IFN-α, microglia upregulate specific TLRs such as TLR3, TLR4, and TLR7 on DCs, NK cells, and lymphocytes, and thus increase the production of IFN and cytokines [253]. Specifically, CCL5, CCL2, and CXCL10 recruit peripheral immune cells to the infected brain [321,322,323], of which CXCL10 in particular has been shown to attract lymphocytes to HSV-1 infection [193]. Microglial IFN-β further mediates the production of IL-10, a known immunosuppressor, which can prevent severe inflammation [324], and IL-6, which protects against neuronal loss [325]. Cytokines and chemokines primarily originating from microglia also feature neurotoxic activity as it has been reported for TNF and IL-1β [193]. HSV-1-infected microglial cells have been shown to produce ROS, leading to higher amounts of inducible nitric oxide synthase and cytotoxic nitric oxide, which are responsible for brain damage [250]. Microglia-mediated immune responses can also have detrimental long-term consequences. Chronic HSV-1 infection leads to persistent activation of microglia [251], resulting in prolonged upregulation of ISGs, reinforcing chronic inflammation and neuronal damage [326]. 

## 6. Future Directions

Even though the CNS remains a highly vulnerable tissue for a few specialized viral diseases, such as rabies and HSE, little is known about the role of glial cells during neuroinvasive infections. Deciphering the underlying mechanisms of highly distinct viral growth characteristics between different CNS cell types may unravel factors that limit viral replication. 

To date, neurotropic RABV is often studied in animal models and neuronal cell types using lab-attenuated viruses. Therefore, data are scarce about the role of non-neuronal CNS cell types during viral encephalomyelitis in humans. Similar to rabies, the roles of astrocytes, oligodendrocytes, and microglia during HSE are still not fully understood. Studying HSV or RABV in neuronal monocultures does not reflect the diversity of the CNS. More importantly, the crosstalk between neurons, astrocytes, oligodendrocytes, and microglia is crucial to ensure CNS homeostasis and to protect the infected CNS from pathological conditions. More sophisticated models, which reflect the human brain in a physiological context, are needed to understand the interaction of HSV and RABV with cell-specific signaling pathways. Thus, determining the molecular mechanisms, which define the cellular susceptibility to neurotropic HSV and RABV infection, remains crucial to discover new and promising therapeutic targets.

## Figures and Tables

**Figure 1 viruses-13-02364-f001:**
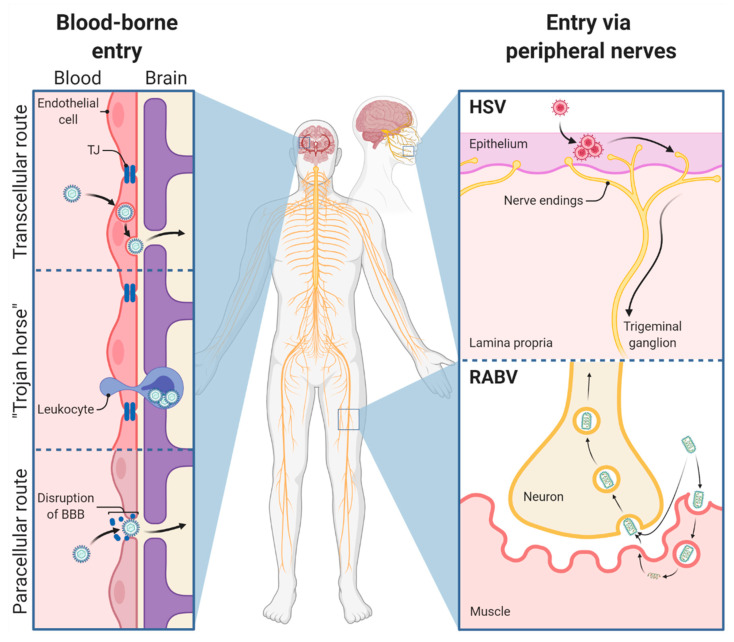
Infection routes of neurotropic viruses. Pathogens can enter the CNS via two distinct routes, the blood (**left**) or the nervous system (**right**). Entry via the bloodstream is mediated by crossing the BBB (**left**) or the blood–cerebrospinal fluid barrier (not illustrated here). The mechanisms for crossing the BBB include the transcellular, ‘Trojan horse’, or paracellular routes [2]. Transcellular transport describes the transport of viruses by a direct diffusion pathway across the endothelial monolayer [5]. The ‘Trojan horse’ route refers to viruses that infect peripheral cells and enter the CNS parenchyma via paracellular or transcellular means. Paracellular entry of viruses is associated with the disruption of the BBB preferably via alterations in the phosphorylation status of tight junction proteins, the disruption of the actin cytoskeleton, or the disruption of the basal lamina [4]. In contrast to this, RABV and HSV bypass the neuroprotective barriers and enter the CNS via retrograde transport along peripheral nerves (**right**). Upon the bite of an infected rabid animal, RABV enters the broken skin and infects peripheral nerves at the motor endplate of neuromuscular junctions or other innervated tissues. After receptor-mediated entry, RABV is retrogradely transported in endosomal transport vesicles to higher-order neurons until the virus reaches the CNS [6]. HSV enters the termini of sensory neurons, which innervate the exposed skin. It is then retrogradely transported to autonomic nerve terminals of neurons in the trigeminal ganglion (TG), in which viral DNA remains in a latency phase. After reactivation, HSV can reach the brain via anterograde transport [7].

**Figure 2 viruses-13-02364-f002:**
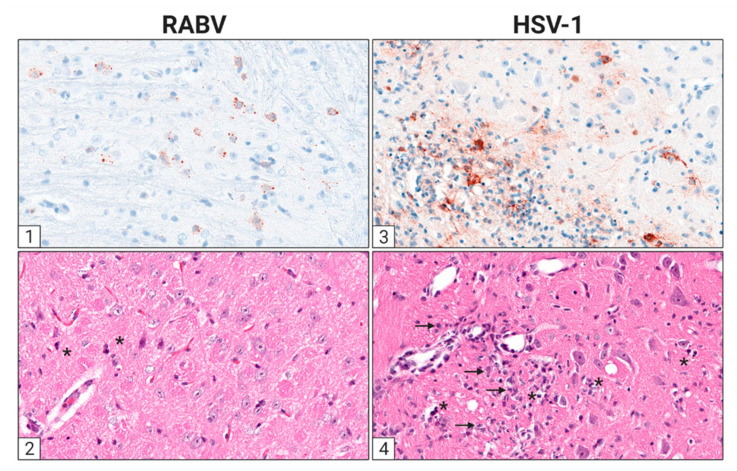
Comparative histopathology of RABV and HSV-1 infection in the murine brain. (**1**) Multiple brain stem neurons show positive signals against RABV antigen. The microscopic picture presents an immunohistochemistry staining using the ABC method. (**2**) Only a small number of necrotic neurons (asterisk) are present in the consecutive section of the RABV-infected brainstem. Inflammatory cells are absent. The microscopic picture presents hematoxylin and eosin staining. (**3**) Multiple neurons and glial cells of a murine brainstem are positive for HSV-1 antigen. The microscopic picture presents immunohistochemistry staining using the ABC method. (**4**) In contrast to RABV, as demonstrated in the consecutive section, HSV-1 infection induces a marked inflammatory response mainly consisting of infiltrating histiocytes and lymphocytes (arrow). Infected cells are necrotic (asterisk). The microscopic picture presents hematoxylin and eosin staining.

**Figure 3 viruses-13-02364-f003:**
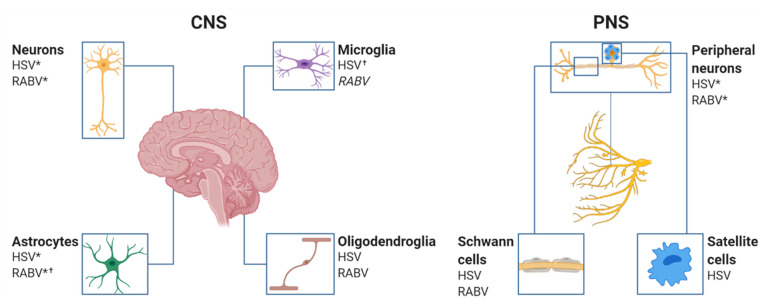
Cellular tropism of RABV and HSV in the CNS and PNS. Both RABV and HSV infect a variety of cells in the CNS (**left**) and PNS (**right**) in both abortive (†) and nonabortive (*) manners. If neither has been demonstrated thus far or productive infection can vary, no indicator is assigned. An abortive infection is defined by the inability of an infected cell to generate infectious progenitor virus, but it can still result in the activation of (innate) immune mechanisms. For RABV, a strain-specific ability of pathogenic RABV to nonabortively infect murine astrocytes in vivo has been demonstrated [122]. Positive viral antigen staining has been demonstrated to occur in neurons, astrocytes, and oligodendrocytes of postmortem human brain tissues [192], whereas infection of SCs has thus far only been shown in a mouse animal model [123]. Fetal human microglia have only stained positive for RABV antigen in vitro (italics) [127]; however, the susceptibility of human induced pluripotent stem cell-derived microglia-like cells has been questioned recently [160]. In postmortem human brain tissues, neurons, astrocytes, and oligodendrocytes have stained positive for HSV antigens [173], whereas infected SCs have only been demonstrated in a mouse animal model [187]. Microglia are supposedly not permissive to HSV infection [182,193].

**Figure 4 viruses-13-02364-f004:**
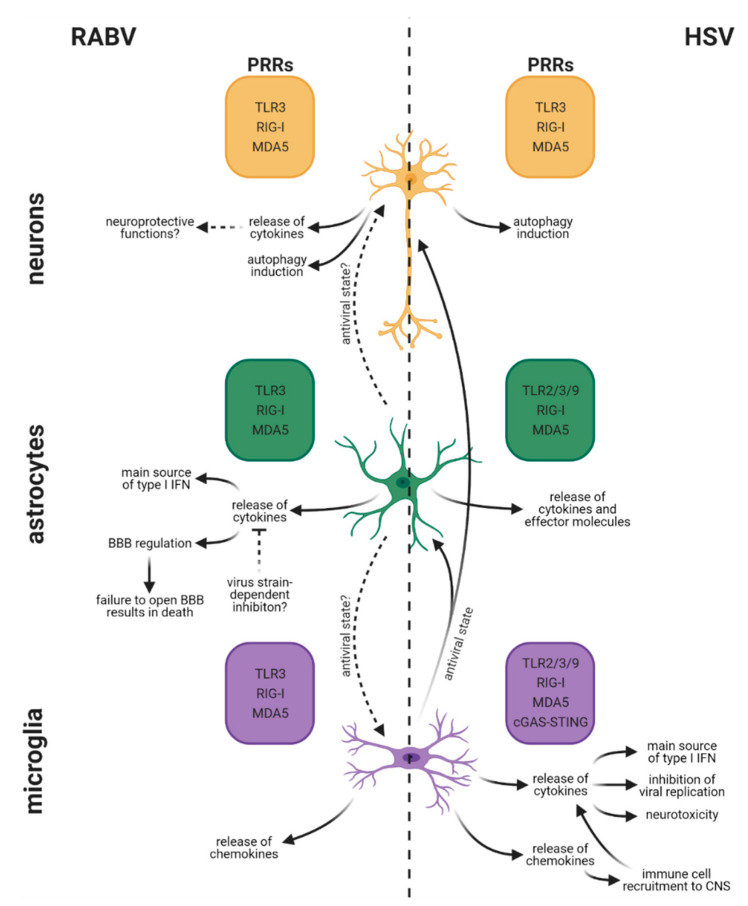
Hypothesis of cell-specific innate immune responses of CNS cell types upon RABV and HSV infection. Neurons (**yellow**), astrocytes (**green**), and microglia (**purple**) are productively or nonproductively infected by HSV and RABV (Figure 3). Infection is sensed via a cell-specific set of PRRs, which trigger downstream signaling pathways and lead to the induction of an innate immune response [239]. Whereas certain PRRs are associated with RABV and HSV infection, neurons do not exhibit evident immune responses upon viral infection. It is suggested that neurons lack robust innate immune signaling to minimize harm and rather require autophagy to restrict RABV [247] and HSV [246,248] replication. In contrast, astrocytes and microglia respond to RABV and HSV infection via the production of cytokines and chemokines [109,144,193,249,250,251,252,253]. While for HSV, microglia are the main source of type I IFN in the CNS and result in pronounced immune cell infiltration [108], astrocytes comprise the primary type I IFN producers for RABV [109]. Dashed arrows indicate putative or suggested effects.

## Data Availability

No new data were created or analyzed in this study. Data sharing is not applicable to this article.

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
