# Peer review of "Innate Immune Signaling and Role of Glial Cells in Herpes Simplex Virus- and Rabies Virus-Induced Encephalitis"

_viruses, 2021, doi:10.3390/v13122364_

Round 1
Reviewer 1 Report
In this review, the authors provide a broad review of the literature regarding the immune responses of resident CNS cell to two clinically important and dissimilar neurotropic viruses, Rabies virus and HSV. In general, the piece is well written and comprehensive, and it is supplemented by effective and well-designed graphics. However, there are a number of issues that should be addressed to improve the work.
MAJOR POINTS
- There is almost no mention of the known/possible role of IL-10 and its related family members in the maintenance of the immune-quiescent environment of the CNS.
- In the first paragraph of page 6, the ability of astrocytes to have antigen presenting functions following activation should also be acknowledged.
- The description of the molecular mechanisms by which resident glial cells perceive these viruses is split between the introduction of cytosolic DNA sensing molecules on page six and the longer description of membrane associated PRRs and cytosolic RNA sensors later on. Most notably, the possible role for DNA sensing PRRs is absent from this later section and Figure 4.
- First paragraph page 16, the known/possible role of other programmed death pathways such as necroptosis and pathanatos is not discussed.
- Line 603, please note that PRR expression is often dependent on the activation and/or differentiation status of the glia.
- Line 765, please be more circumspect with regard to the relative importance of TLR3 in HSV-1 as the work cited (ref 288) is now almost a decade old.Indeed, a number of more recent studies have indicated an important role for cytosolic nucleic acid sensors such as RIG-I and ZBP1 (aka DAI). This work is not discussed in the present manuscript. Furthermore, no mention is made of the possible detection of viral DNA via RNA sensing PRRs due to the activity of Pol III.
MINOR POINTS
- Syntactical and/or grammatical errors lines 26, 42, 95, 104, 116, 123, 129, 133, 137, 140, 172, 177,180, 193, 204, 207, 218, 219, 222, 229, 231, 285, 288, 296, 298, 305, 312, 322, 323, 358, 381, 382, 411, 435, 468, 470, 510, 514, 517, 519, 521, 527, 541, 559, 563, 568, 570, 571/572, 574, 583588, 589, 590, 652, 674, 707, 737, 772, 834, 836, and 842.
- Typographical errors lines 389 and 439
- Line 43, please note additional involvement of innate immune cells.
- Please add paragraph breaks lines 93, 142, 527, 572, 586, and 704
- Line 227, please note that, while microglia exhibit many functional similarities to macrophages, they are different cell types.
- Line 251 change to “La Crosse”.
- Please use complete sentences in the legend for Figure 2.
- Please be consistent in nomenclature use for MAVS/IPS-1.
- On several occasions rabies viruses are referred to as “street” strains, is “field” intended?
- Line 329 is “highly” intended rather than “increasingly”?
- Line 457 change to “notion that these cells (i) are not”.
- It is suggested that the authors use differently colored fonts for abortive versus non-abortive in Figure 3 in addition to the symbols.
- Line 550, please delete first “consequently”
- Line 558, please note that CCL2 and CCL5 are also chemokines.
- Please note that the naming convention has changed such that TNF rather than TNF-alpha is used as TNF-beta has been renamed lymphotoxin-alpha.
Author Response
Please find attached the point-by-point responses. Thank you for your time to read and improve the manuscript.
Kind regards,
Lena Feige

Reviewer 2 Report
This is a very thorough review of the interactions that HSV1 and RABV have with CNS cell types, especially focused on how glial cell lineages mediate detection and immune responses leading to dichotomous pathologies.
I have no major comments, only a few minor issues and grammatical suggestions:
Line239: The upregulation, or even expression of MHC class II on microglia can be controversial. The authors should make sure to cite primary literature that shows appreciable MHC Class II upregulation on primary microglia in vivo.
Line 439: IFNs is written INFs
Line 514: Increased permeable BBB doesnt read like proper english. Maybe "increasing BBB permeability"
Line 835: Data ARE scarce...not is.
Author Response

(The authors gave the same response as above.)

Round 2
Reviewer 1 Report
The authors have addressed the concerns raised to my satisfaction.